# SHIFT-RESILIENT DIFFUSIVE IMPUTATION FOR VARIABLE SUBSET FORECASTING

## ABSTRACT

It is common for sensor failures to result in missing data, leading to training sets being complete while test sets have only a small subset of variables. The challenge lies in utilizing incomplete data for forecasting, which is known as the Variable Subset Forecasting (VSF). In VSF tasks, significant distribution shift is present. One type is inter-series shift, which indicates changes in correlations between different series, and the other type is intra-series shift, which refers to substantial distribution differences within the same series across different time windows. Existing approaches to solving VSF tasks typically involve imputing the missing data first and then making predictions using the completed series. However, these methods do not account for the shift inherent in VSF tasks, resulting in poor model performance. To address these challenges, we propose a **S**hift-**R**esilient **D**iffusive **I**mputation (SRDI) framework against the shift. Specifically, SRDI integrates divide-conquer strategy with the denoising process, that decomposes the input into invariant patterns and variant patterns, representing the temporally stable parts of inter-series correlation and the highly fluctuating parts, respectively. By extracting spatiotemporal features from each separately and then appropriately combining them, inter-series shift can be effectively mitigated. Then, we innovatively organize SRDI and the forecasting model into a meta-learning paradigm tailored for VSF scenarios. We address the intra-series shift by treating time windows as tasks during training and employing an adaptation process before testing. Extensive experiments on four datasets have demonstrated our superior performance compared with state-of-the-art methods. Code is available at the repository: https://anonymous.4open.science/r/SRDI-944C.

## 1 INTRODUCTION

In real-world IoT applications, sensor malfunctions and data collection issues often result in missing data in time series, complicating predictive modeling and impairing forecasting performance. A particularly difficult situation arises when entire sequences of data are missing. For instance, a model trained with $N$ variables to predict air quality in one region may need to be deployed in another region with only $S$ ($S \ll N$) available sensors. Additionally, extreme weather conditions can cause sensor damage, leading to incomplete variable recordings in subsequent times. This scenario, known as **Variable Subset Forecasting** (VSF) Chauhan et al. (2022), requires making predictions with only a subset of the variables used during training, which poses significant challenges for achieving accurate forecasts.

One of the most intuitive solutions is to impute the missing variables before making predictions. Numerous imputation methods have been proposed, including recent advancements in diffusion models Tashiro et al. (2021). However, these approaches consistently face significant challenges in VSF scenarios, primarily due to distribution shift that is prevalent in these settings. Specifically, we categorize the distribution shift encountered in VSF into two main types: **(i) Inter-Series Shift**: In VSF scenarios, the absence of variables disrupts the ability to accurately capture relationships between variables. Additionally, the correlations between variables may change unpredictably over time, *i.e.*, covariate shift, leading to systemic inaccuracies in learning these relationships Schneider et al. (2020). This variability significantly degrades the model's performance as it fails to adapt to shifting inter-variable dynamics. **(ii) Intra-Series Shift**: Data in time series forecasting tasks is typically segmented into time windows, each with its own distinct distribution Fan et al. (2023). These distributions may change abruptly over time, rendering the model trained on past data ineffective for new, unseen distributions. This intra-series shift poses a substantial challenge to the imputation model's generalization ability across varying data distributions. Given these two types of shift, existing imputation methods prove inadequate for sustaining robust performance in VSF environments.

Our objective is to develop a robust imputation model that effectively handles both inter-series and intra-series shift, ensuring satisfying performance for VSF.

To address the above challenges, we propose a shift-resilient diffusive imputation framework for VSF. Specifically, we outline our solutions against the two types of shift as follows.

To effectively manage inter-series shift in VSF, our approach integrates Divide-Conquer strategy during the denoising process in the diffusive imputation. This technique involves decomposing the time series data into two distinct patterns: invariant and variant. The invariant pattern focuses on capturing the stable, underlying correlations that do not change significantly over time, providing a robust foundation for the model. In contrast, the variant pattern addresses the dynamic correlations that are susceptible to shift. The decomposition allows the model to specifically target and adapt to changes in variable relationships, enhancing its ability to accurately impute missing data amidst evolving conditions. By processing these patterns separately and then recombining them, our model effectively isolates and compensates for the variability caused by inter-series shift, thus maintaining high accuracy in variable recovery.

For intra-series shift, which occurs due to abrupt changes in data distribution within the same series over different time windows, we employ a meta-learning strategy within our diffusive imputation framework. This strategy trains the model to rapidly adapt to new distributions by treating imputation over each time window as a distinct task. Meta-learning enables the diffusive imputation framework to learn from a variety of distribution scenarios, enhancing its flexibility and generalization capability. By continuously updating its parameters in response to new data characteristics, the model is equipped to handle previously unseen distributions effectively. This adaptive capability is critical for maintaining consistent imputation performance across varying data landscapes, particularly in VSF environments where the model must reconstruct the missing variables accurately despite significant shift in data distribution.

In summary, our contributions can be summarized as follows:

- We introduce a novel diffusive imputation method specifically designed for Variable Subset Forecasting (VSF) tasks, marking the first known application in this context.
- We categorize and provide a comprehensive analysis of two distinct types of shift prevalent in VSF tasks: inter-series and intra-series shift.
- We develop a divide-conquer denoising model tailored for effectively addressing inter-series shift, alongside a meta-learning strategy that enhances the model's adaptability to intra-series shift.
- We validate our approach through extensive experimentation on four real-world datasets, demonstrating consistent superiority in effectiveness compared with existing state-of-the-art methods tailored to VSF tasks.

## 2 RELATED WORK

### 2.1 TIME SERIES IMPUTATION TECHNOLOGIES

Time series imputation fills missing time points in a series and can be categorized into simple and machine learning-based methods Luo et al. (2018). Early approaches, like mean, median, or mode imputation Donders et al. (2006); Acuna & Rodriguez (2004); Kantardzic (2011), were later surpassed by machine learning-based methods, such as K-Nearest Neighbors MATLAB & Release (2019) and neural models like LSTM and CNN Ahn et al. (2022). For Variable Subset Forecasting (VSF), Forecast Distance Weighting (FDW) Chauhan et al. (2022) has shown promise. However, current methods struggle with time series shifts, which degrade performance. Our model addresses this issue, excelling in VSF under challenging conditions.

### 2.2 TIME SERIES DIFFUSION MODEL

Diffusion models are powerful generative tools with remarkable performance across domains. In time series, DCRNN Li et al. (2017) introduced diffusion convolution with recurrent networks to model spatial dependencies for traffic flow prediction. A recent review Lin et al. (2023) summarized

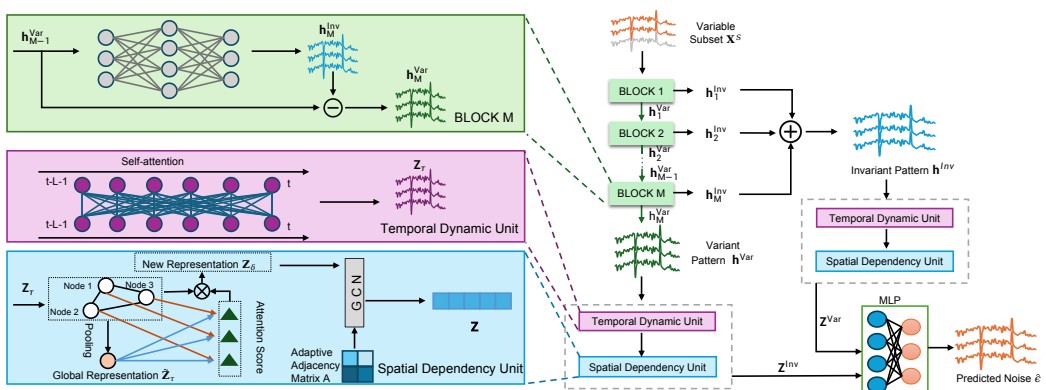

Figure 1: Framework Overview.

their strengths and applications in forecasting. Diffusion models have also proven to be highly effective in time series imputation, CSDI Tashiro et al. (2021) became the first study to apply the diffusion model to time series imputation, achieving significant results. PRISTI Liu et al. (2023) improved upon CSDI, enhancing the effectiveness of imputation. However, their application to the VSF task remains underexplored.

## 3 PROBLEM FORMULATION

Let $\Gamma_N$ denote a N-variate space, $\mathbf{x}_t^N = \{x_t^{(1)}, \cdots, x_t^{(i)}, \cdots, x_t^{(N)}\}$ represent the observations of a $N$-variate time series at the time step $t$, where $x_t^{(i)} \in \Gamma_N$. Then, the $L$-length lookback window can be denoted as $\mathbf{X}^N = \{\mathbf{x}_{t-L-1}^N, \cdots, \mathbf{x}_t^N\}$, and the subsequent $H$-length horizon window is $\mathbf{Y}^N = \{\mathbf{x}_{t+1}^N, \cdots, \mathbf{x}_{t+H}^N\}$.

A small variable subset $\Gamma_S$ ($\Gamma_S \subset \Gamma_N$ and $|\Gamma_S| \ll |\Gamma_N|$) is randomly sampled from $\Gamma_N$. The corresponding lookback window and horizon window observations can be denoted as $\mathbf{X}^S$ and $\mathbf{Y}^S$, respectively. VSF refers to adapting a forecasting model $\mathcal{F}_\Theta$ (parameterized by $\Theta$) trained on the complete observations ($\mathbf{X}^N, \mathbf{Y}^N$) to a variable subset ($\mathbf{X}^S, \mathbf{Y}^S$). During the process, an imputation model is required to recover the missing variables to comply with $N$-variable input forecasting model $\mathcal{F}_\Theta$. The objective is to optimize the forecasting performance on the subset ($\mathbf{X}^S, \mathbf{Y}^S$). Let $\mathcal{F}_\Phi$ denote the imputation model parameterized by $\Phi$, Variable Subset Forecasting task can be represented as

$$\min_\Phi |\mathbf{Y}^S - \Delta_S[\mathcal{F}_\Theta(\mathcal{F}_\Phi(\mathbf{X}^S))]|, \tag{1}$$

where $\Delta_S[\cdot]$ is an indexing function to select the results corresponding to the variable subset $\Gamma_S$.

## 4 SHIFT-RESILIENT DIFFUSIVE IMPUTATION

In this section, we present our diffusive imputation framework, which is shown in Figure 1. The diffusive imputation model leverages noise to impute the missing variables. The presence of inter- and intra-variable shift significantly sharpens the imputation performance. Specifically, to address the impact of the inter-variable shift on model performance, we decompose the time series into invariant and variant patterns as detailed in Section 4.2.1. In Section 4.2.2, we designed a technique for preserving spatiotemporal relations. The extracted invariant and variant patterns are then reasonably combined, which will be introduced in Section 4.2.3, to generate the final output of the diffusive imputation model. Additionally, to mitigate the intra-variable shift, we propose a meta-learning framework, further elaborated in Section 5.

### 4.1 AN OVERVIEW OF CONDITIONAL DIFFUSIVE IMPUTATION WITH VARIABLE SUBSETS

Inspired by CSDI Tashiro et al. (2021), we formulate variable subset imputation as a conditional diffusion process, where variable subset $\mathbf{X}^S$ is considered as the condition to generate the target

missing variables $\mathbf{X}^{N/S}$. To make it consistent in this paper, we set the complete observations $\mathbf{X}^N$, variable subset $\mathbf{X}^S$, and the missing variables $\mathbf{X}^{N/S}$ as the same size, where $\mathbf{X}^S$ is derived by masking the missing variables in $\mathbf{X}^N$, and $\mathbf{X}^{N/S}$ is derived by masking the observed variable subset in $\mathbf{X}^N$, respectively. Thus, $\mathbf{X}^N = \mathbf{X}^S + \mathbf{X}^{N/S}$.

Specifically, the conditional diffusive imputation consists of two phases: the noise-adding phase and the denoising phase. In the noise-adding phase, Gaussian noise is kept added over the missing variables $\mathbf{X}^{N/S}$ iteratively to convert the $\mathbf{X}^{N/S}$ into Gaussian noise:

$$
\begin{aligned}
q(\mathbf{X}_{1:R}^{N/S}|\mathbf{X}^{N/S}) &:= \prod_{r=1}^{R} q(\mathbf{X}_r^{N/S}|\mathbf{X}_{r-1}^{N/S}), \\
q(\mathbf{X}_r^{N/S}|\mathbf{X}_{r-1}^{N/S}) &:= \mathcal{N}(\sqrt{1-\beta_r}\mathbf{X}_{r-1}^{N/S}, \beta_r\mathbf{I}),
\end{aligned}
\tag{2}
$$

where $R$ denotes the total rounds of the noise-adding, $q$ represents the data distribution, $\mathbf{I}$ represents identity matrix, and $\mathcal{N}$ represents Gaussian distribution. $\mathbf{X}_r^{N/S} = \sqrt{\widetilde{\alpha_r}}\mathbf{X}^{N/S} + \sqrt{1-\widetilde{\alpha_r}}\epsilon$, where $\alpha_r = 1 - \beta_r$, $\widetilde{\alpha_r} = \prod_{r=1}^{R}\alpha_r$, $\epsilon$ is the sampled standard Gaussian noise and $\beta_r$ represents the noise level.

The denoising phase represents a reverse process of adding noise. Given an input $\mathbf{X}_R^{N/S}$ that is filled with Gaussian noise, after $R$ denoising steps, the output will be the original, noise-free data $\mathbf{X}^{N/S}$. The denoising phase can be represented as follows:

$$
\begin{aligned}
p_\Phi(\mathbf{X}_{0:R-1}^{N/S}|\mathbf{X}_R^{N/S},\mathbf{X}^S) &:= \prod_{r=1}^{R} p_\Phi(\mathbf{X}_{r-1}^{N/S}|\mathbf{X}_r^{N/S},\mathbf{X}^S), \\
p_\Phi(\mathbf{X}_{r-1}^{N/S}|\mathbf{X}_r^{N/S},\mathbf{X}^S) &:= \mathcal{N}(\mathbf{X}_{r-1}^{N/S};\mu_\Phi(\mathbf{X}_r^{N/S},\mathbf{X}^S,r),\sigma_r^2\mathbf{I}),
\end{aligned}
\tag{3}
$$

where $\mu_\Phi(\mathbf{X}_r^{N/S},\mathbf{X}^S,r) = \frac{1}{\sqrt{\widetilde{\alpha_r}}}(\mathbf{X}_r - \frac{\beta_r}{\sqrt{1-\widetilde{\alpha_r}}}\epsilon_\Phi(\mathbf{X}_r^{N/S},\mathbf{X}^S,r))$, $\sigma_r^2 = \frac{1-\widetilde{\alpha}_{r-1}}{1-\widetilde{\alpha}_r}\beta_r$, $\epsilon_\Phi$ represents a trainable denoising function parameterized by $\Phi$, and $\mathbf{X}_0^{N/S} = \mathbf{X}^{N/S}$ denotes the recovered clean missing variables from noise. Then, the learning objective is to optimize the following loss function:

$$
\mathcal{L}^{\text{diff}} = E_{\mathbf{X}^{N/S}\sim q(\mathbf{X}^{N/S}),\epsilon\sim\mathbf{N}(0,I)}\|\epsilon - \epsilon_\Phi(\mathbf{X}_r^{N/S},\mathbf{X}^S,r)\|^2.
\tag{4}
$$

As suggested by PriSTI Liu et al. (2023), the denoising function $\epsilon_\Phi$ is inherently a noise prediction function. Therefore, the conditional diffusive model learns the variable imputation capability by predicting the added noise, and thus recover the missing variables $\mathbf{X}^{N/S}$. Specifically, during the imputation process, the input is a time series with missing variables, *i.e.*, variable subset $\mathbf{X}^S$, where the missing parts are represented as empty (naturally masked due to unavailability). We directly fill the missing variables with Gaussian noises to convert the missing variables into Gaussian noises, thus obtaining $\mathbf{X}_1^{N/S}$, *i.e.*, $\mathbf{X}_R^{N/S}$ where $R = 1$. Then, we take $(\mathbf{X}_1^{N/S},\mathbf{X}^S)$ as input to the well-trained denoising function, $\epsilon_\Phi$, to derive the conditional probability $p_\Phi(\mathbf{X}_0^{N/S}|\mathbf{X}_1^{N/S},\mathbf{X}^S)$, *i.e.*, $p_\Phi(\mathbf{X}^{N/S}|\mathbf{X}_1^{N/S},\mathbf{X}^S)$ according to Equation 3. Finally, the imputed data $\mathbf{X}^{N/S}$ can be derived by sampling from $p_\Phi(\mathbf{X}^{N/S}|\mathbf{X}_1^{N/S},\mathbf{X}^S)$, *i.e.*, $\mathbf{X}^{N/S} \sim p_\Phi(\mathbf{X}^{N/S}|\mathbf{X}_1^{N/S},\mathbf{X}^S)$.

It is evident that the success of the diffusion model in imputation depends on the rational design of the denoise function Tashiro et al. (2021); Liu et al. (2023). Therefore, we design the denoising function in a divide-conquer manner to facilitate the diffusive imputation model with the capability of addressing the inter-series shift. Specifically, we decompose the complicated and nested parts into the invariant pattern (parts with relatively stable inter-variable correlations) and the variant pattern (parts with inter-variable correlation changes). By learning spatiotemporal characteristics separately for these components and then integrating them, we can effectively mitigate the interference of inter-series shift, thereby enhancing the performance of the variable imputation. We introduce the detailed design of the divide-conquer denoising function in the following section.

## 4.2 DIVIDE-CONQUER DENOISING

### 4.2.1 DISENTANGLING INVARIANT-VARIANT PATTERNS

We introduce "Invariant-variant Dispatcher" for distangling invariant and variant patterns. The dispatcher is composed of $M$ blocks Oreshkin et al. (2019), and blocks are stacked and collaboratively contribute to distangling invariant and variant patterns. For a general description, we take the $m$-th block for illustration. Formally, let $\mathbf{h}_{m-1}^{\text{Var}}$ denote the learned variant patterns from the $(m-1)$-th block. The $m$-th block takes the variant patterns $\mathbf{h}_{m-1}^{\text{Var}}$ as input, and further refines it into more specific invariant patterns $\mathbf{h}_m^{\text{Inv}}$ and variant patterns $\mathbf{h}_m^{\text{Var}}$. Thus, the $m$-th block can be represented as

$$
\begin{aligned}
\mathbf{h}_m^{\text{Inv}} &= \text{MLP}_m(\mathbf{h}_{m-1}^{\text{Var}}), \\
\mathbf{h}_m^{\text{Var}} &= \mathbf{h}_{m-1}^{\text{Var}} - \mathbf{h}_m^{\text{Inv}}.
\end{aligned}
\tag{5}
$$

where $\text{MLP}_m$ denotes the multi-layer percetron at the $m$-th block, and $\mathbf{h}_{m-1}^{\text{Var}}$ and $\mathbf{h}_m^{\text{Var}}$ are in the same size. We set the input of the first block as the noisy missing variables $\mathbf{X}_R^{N/S}$.

To ensure $\text{MLP}_m$ indeed learns the invariant patterns, we constrain the correlation disparity between the consecutive time steps to be as small as possible. Let $\mathbf{C}_t$ and $\mathbf{C}_{t+1}$ denote the the correlation matrix of $\mathbf{h}_m^{\text{Inv}}$ at the time step $t$ and $t+1$, respectively. Then, we implement the constraint by minimizing the following loss function

$$
\mathcal{L}_m^{\text{disp}} = \sum_{t=1}^{T-1} ||\mathbf{C}_{t+1} - \mathbf{C}_t||_2^2.
\tag{6}
$$

Due to the page limitation, we introduce the calculation of the correlation matrix $\mathbf{C}_t$ in A.5.

Then, we accumulate the invariant patterns from all $M$ blocks as the final invariant patterns $\mathbf{h}^{\text{Inv}}$, and take the derived variant patterns from the last block as the final variant patterns $\mathbf{h}^{\text{Var}}$:

$$
\begin{aligned}
\mathbf{h}^{\text{Inv}} &= \sum_{m=1}^{M} \mathbf{h}_m^{\text{Inv}}, \\
\mathbf{h}^{\text{Var}} &= \mathbf{h}_M^{\text{Var}}.
\end{aligned}
\tag{7}
$$

Additionally, for the entire dispatcher, we accumulate all the correlation disparity loss functions to serve as regularization, ensuring invariant pattern learning:

$$
\mathcal{L}^{\text{disp}} = \sum_{m=1}^{M} \mathcal{L}_m^{\text{disp}}.
\tag{8}
$$

Through the continual refinement and collaboration by the $M$-block dispatcher, the generated invariant patterns and variant patterns can be effectively disentangled.

### 4.2.2 PRESERVING SPATIOTEMPORAL CHARATERISTICS

After disentangling invariant and variant patterns, we proceed to capture the spatiotemporal characteristics of each branch. Specifically, we develop the Temporal-Spatial Representation (TSR) Module, which consists of the Temporal Dynamic Unit and the Spatial Dependency Unit. The invariant and variant patterns exploit the same TSR module architecture but learn the parameters separately. To avoid redundancy, we represent the disentangled pattern as $\mathbf{h}$ and omit the subscripts "Inv" and "Var" in the following description.

**Temporal Dynamic Unit.** We exploit self-attention mechanisms to encode the temporal dynamics of each time step. Therefore, we can obtain the temporally-learned representation $\mathbf{Z}_\tau$, which will serve as the input for the Spatial Dependency Unit.

$$
\mathbf{Z}_\tau = \text{SoftMax}(\frac{\mathbf{W}_\tau^q(\mathbf{h})\mathbf{W}_\tau^k(\mathbf{h})}{\sqrt{d_\tau}})\mathbf{W}_\tau^v(\mathbf{h}),
\tag{9}
$$

where $d_\tau$ denotes the hidden dimension; $\mathbf{W}_\tau^q$, $\mathbf{W}_\tau^k$, and $\mathbf{W}_\tau^v$ represent the learnable weight matrix corresponding to the query, key, and value, respectively.

**Spatial Dependency Unit.** We formulate the dependencies between variables from a graph perspective, where each node denotes one variable, the edge demonstrates the dependencies between variables, and the learned temporal representations $\mathbf{Z}_\tau$ are the initial node features. For a unified representation in the spatial scope, we first calibrate node representations through a global-local attention mechanism. Specifically, we perform a graph pooling operation on $\mathbf{Z}_\tau \in \mathbb{R}^{T \times N}$ to obtain $\tilde{\mathbf{Z}}_\tau \in \mathbb{R}^{T \times 1}$, which represents a global representation encapsulating information from all variables:

$$\tilde{\mathbf{Z}}_\tau = \mathrm{Pooling}(\mathbf{Z}_\tau). \tag{10}$$

Then, we calculate the attention scores between the global representation $\tilde{\mathbf{Z}}_\tau$ and each node in $\mathbf{Z}_\tau$. We leverage these global-local attention scores to calibrate the representation as

$$\mathbf{Z}_\delta = \mathrm{SoftMax}\left(\frac{\mathbf{W}_\delta^q(\tilde{\mathbf{Z}}_\tau)\mathbf{W}_\delta^k(\mathbf{Z}_\tau)}{\sqrt{d_\delta}}\right)\mathbf{W}_\delta^v(\mathbf{Z}_\tau), \tag{11}$$

where $d_\delta$ denotes the hidden dimension; $\mathbf{W}_\delta^q$, $\mathbf{W}_\delta^k$, and $\mathbf{W}_\delta^v$ represent the learnable weight matrix corresponding to the query, key, and value, respectively.

Moreover, we employ an adaptive Graph Convolutional Network (GCN) BAI et al. (2020) to learn spatial dependencies. We first initialize a learnable embedding $\mathbf{E} \in \mathbb{R}^{N \times d_n}$, with $d_n$ hidden dimension, to reconstruct an adaptive adjacency matrix $\mathbf{A}$

$$\mathbf{A} = \mathrm{SoftMax}(\mathrm{ReLU}(\mathbf{E}\mathbf{E}^T)). \tag{12}$$

The spatiotemporal representations $\mathbf{Z}$ can be calculated with the massage passing mechanism Zhao et al. (2020a):

$$\mathbf{Z} = \mathbf{A}\mathbf{Z}_\delta\mathbf{W}, \tag{13}$$

where $\mathbf{W}$ denotes the learnable weight matrix. We follow the same pipeline to generate the invariant pattern representations $\mathbf{Z}^{\mathrm{Inv}}$ and the variant pattern representations $\mathbf{Z}^{\mathrm{Var}}$, respectively.

### 4.2.3 FUSING NOISE PREDICTION

After separately learning the representations of invariant and variant patterns, we concatenate $\mathbf{Z}^{\mathrm{Inv}}$ and $\mathbf{Z}^{\mathrm{Var}}$ for the final noise prediction

$$\hat{\epsilon} = \mathrm{MLP}(\mathbf{Z}^{\mathrm{Inv}} \,||\, \mathbf{Z}^{\mathrm{Var}}), \tag{14}$$

where $\hat{\epsilon}$ denotes the predicted noise that can also be represented as

$$\hat{\epsilon} = \epsilon_\Phi(\mathbf{X}_r^{N/S}, \mathbf{X}^S, r). \tag{15}$$

We substitute Equation 14 and Equation 15 to Equation 4 for calculating the diffusion loss $\mathcal{L}^{\mathrm{diff}}$.

Therefore, considering the invariant-variant disentanglement, the Divide-Conquer Denoising (DCD) loss can be represented as

$$\mathcal{L}^{\mathrm{DCD}} = \mathcal{L}^{\mathrm{diff}} + \varpi \cdot \mathcal{L}^{\mathrm{disp}}, \tag{16}$$

where $\varpi$ is a hyperparameter to control the contribution of $\mathcal{L}_{\mathrm{disp}}$.

## 5 META LEARNING STRATEGY AGAINST INTRA-SERIES SHIFT

In this section, we introduce a meta-learning strategy to eliminate the intra-series shift. We divide the time series into multiple windows, treating each window as a separate task. By the inner-outer loop of training the model parameters across these different tasks, we aim to ensure that the trained model can effectively adapt to tasks across different time windows, thereby addressing intra-series shift. Specifically, the proposed meta-learning strategy mainly includes two stages: *Stage 1, the meta-training stage:* we optimize the initial parameters through the learning of multiple diffusion models followed by a forecasting backbone, enabling rapid adaptation to the inference phase. *Stage 2, the adaptation stage:* we use the variable subset in the inference phase to quickly adjust the initial meta-model parameters, enabling it to adapt to and address the variable subset forecasting task. Next, we introduce the two stages in detail.

---

**Algorithm 1** Meta-Training Stage

---

**Require:** $p(k)$: distribution over windows(tasks)
**Require:** $\eta, \gamma$ :learning rate
1: randomly initialize $\Theta, \Phi$
2: **while** not done **do**
3:     Sample batch of tasks $k \sim p(k)$
4:     **for** all $k$ **do**
5:         Evaluate $\nabla_\Phi \mathcal{L}_k^{\text{DCD}}$ with respect to $K$ examples
6:         Compute adapted parameters of diffusion model with gradient descent and update: $\Phi_k \leftarrow \Phi - \eta \cdot \nabla_\Phi \mathcal{L}_k^{\text{DCD}}$
7:         Do inference and compute the forecasting loss $\mathcal{L}_k^{\text{fcst}}$
8:     **end for**
9:     Jointly update the diffusion model and forecasting model $\Phi \leftarrow \Phi - \gamma \cdot \nabla_\Phi \sum_{k \in \mathbf{K}}(\mathcal{L}_k^{\text{DCD}} + \mathcal{L}_k^{\text{fcst}})$ ;
    $\Theta \leftarrow \Theta - \gamma \cdot \nabla_\Theta \sum_{k \in \mathbf{K}} \mathcal{L}_k^{\text{fcst}}$
10: **end while**

---

### 5.1 META-TRAINING STAGE

The meta-training stage is divided into two parts: the inner loop and the outer loop, which are responsible for rapid adaptation and global optimization of the model, respectively. The full algorithm is outlined in Algorithm 1.

**Inner Loop** We take imputation and forecasting for each $L$-length time window as a task. Specifically, for the $k$-th task, the corresponding parameters set and the denoising loss can be denoted as $\Phi_k$ and $\mathcal{L}_k^{\text{DCD}}$ respectively. Then, the parameter update is represented as

$$\Phi_k \leftarrow \Phi - \eta \cdot \nabla_\Phi \mathcal{L}_k^{\text{DCD}}, \tag{17}$$

where $\eta$ is the learning rate. We iterate all the tasks to update the diffusive imputation model and forecasting model for each respective task.

**Outer Loop** For each task, we leverage the updated diffusive imputation model to generate the missing data, and apply the imputed data to train a forecasting model. Note that all the imputation tasks share the same forecasting model. Let $\mathcal{L}_k^{\text{fcst}}$ denote the forecasting loss on the $k$-th imputed data. Then, we update the meta-model for the diffusive imputation and forecasting model simultaneously as

$$\Phi \leftarrow \Phi - \gamma \cdot \nabla_\Phi \sum_{k \in \mathbf{K}}(\mathcal{L}_k^{\text{DCD}} + \mathcal{L}_k^{\text{fcst}}), \tag{18}$$

$$\Theta \leftarrow \Theta - \gamma \cdot \nabla_\Theta \sum_{k \in \mathbf{K}} \mathcal{L}_k^{\text{fcst}}, \tag{19}$$

where $\gamma$ is a learning rate.

### 5.2 META ADAPTION STAGE

#### 5.2.1 FINE-TUNING

Given a new variable subset, we aim to apply the trained imputation model and forecasting model for the inference. Considering the new variable subset as a new task, we will first conduct fine-tuning for the trained diffusive imputation model following the convention of the meta learning paradigm. In other words, the imputation model requires several iterations of training on the new subset. Unfortunately, due to the existence of missing variables, where the ground truth is unavailable, it is impractical to re-conduct the original training pipeline. To address the issue, we temporally ignore the missing variables, but randomly select pseudo-missing variables from the available subset. We will take the newly selected pseudo-missing variables as the imputation target (with ground truth), and re-launch the inner-loop training pipeline indicated in Equation 17. During the process, the forecasting model is fixed and no longer updated.

### 5.2.2 INFERENCE

After fine-tuning, we shift the focus to the real missing variables as the target and the original available subset as the condition. We then apply the fine-tuned diffusive imputation model to impute the missing variables. This process is described by Equation 3, where the denoise function is known, allowing us to easily obtain the final imputed data. Next, the imputed data is fed into the forecasting model to generate the final predicted values.

## 6 EXPERIMENTS

### 6.1 EXPERIMENTAL SETUP

**Datasets.** We conducted experiments on four datasets: 1) METR-LA; 2) SOLAR; 3) TRAFFIC; 4) ECG5000. For more details on the datasets, please refer to A.4.

**Time Series Forecasting Model Backbone Setting.** Our imputation model can be applied to multiple forecasting models. In our experiments, we integrated it with four commonly used forecasting models: MTGNN Wu et al. (2020), ASTGCN Guo et al. (2019), MSTGCN Jia et al. (2021) , and T-GCN Zhao et al. (2020b). For a detailed description of the backbones, please refer to A.2. In the context of VSF, the data is fully available during training, whereas only a limited subset is accessible during testing. To enhance the accuracy of forecasting results, we initially perform data imputation before feeding the data into the trained forecasting model to generate predictions. To validate the effectiveness of our model, we consider the following two scenarios: **1) Partial**: In this scenario, we utilize only the $N - S$ variables for prediction without performing any imputation of missing values. The resulting prediction outcomes thus represent the inherent performance of the forecasting model in VSF problems. **2) Oracle:** This is a comparative experiment that represents an idealized scenario, seldom observed in practice, where all $N$ variables are fully known. In this case, we use all available variables for forecasting and compute the resulting prediction error.

**Evaluation Metrics.** We assess the performance using two commonly employed metrics in multivariate time series forecasting: Mean Absolute Error (MAE) and Root Mean Squared Error (RMSE). To demonstrate the improvement of our model in the partial setting, we calculated the improvement ratio: $Improved$. For detailed information of the metrics, please refer to A.1.

**Implementation.** We employed the PyTorch framework to implement our model and baselines, and the models were evaluated on a Linux server with a single GPU. We utilized MAE (Mean Absolute Errors) as the loss function During the testing phase, we only had knowledge of a subset $S$. In the main experiment, we selected 15% of the variables to form subset $S$. During the training phase, to demonstrate the reliability of our model, we randomly constructed the subset 100 times to cover as much of the dataset as possible, and trained for 100 epochs. We computed the mean and standard deviation of the models, based on the results. The forecasting horizon length, denoted as $H$, was set to 12, and the lookback window length, denoted as $L$, was also set to 12. In terms of dataset segmentation, 70% of the samples were allocated for training, 10% for validation, and 20% for testing. For further details on hyperparameter settings, please refer to A.6.

### 6.2 OVERALL PERFORMANCE

**Comparison with Partial & Oracle Settings.** Table 1 presents the experimental results of our model using four different backbones across four datasets. The results show that, in the partial setting, compared to the results without imputation, our model achieved average MAE improvements of 20.62%, 12.38%, 32.75%, and 18.87% on the METR-LA, TRAFFIC, SOLAR, and ECG5000 datasets, respectively, demonstrating the effectiveness of SRDI. Moreover, in most datasets, our model even outperforms the oracle, which can be attributed to our successful handling of the interference caused by distribution shift.

**Comparison with Imputation Methods.** To validate the reliability of the proposed Shift-Resilient Diffusive Imputation method, we conducted a comparative analysis with several state-of-the-art imputation models known for their excellent performance: MICE Van Buuren & Groothuis-Oudshoorn (2011), IIM Zhang et al. (2019), TRMF Yu et al. (2016), CSDI Tashiro et al. (2021), FDW Chauhan et al. (2022), SSGAN Miao et al. (2021), TRF Hu et al. (2024), PRISTI Liu et al. (2023, GINAR Yu

Table 1: Comparison with Partial and Oracle settings regarding different forecasting backbones.

| Models | | METR-LA | | TRAFFIC | | SOLAR | | ECG5000 | |
|---|---|---|---|---|---|---|---|---|---|
| | | MAE | RMSE | MAE | RMSE | MAE | RMSE | MAE | RMSE |
| MTGNN | Partial | 4.54(0.37) | 8.90(0.68) | 18.57(2.31) | 38.46(3.94) | 4.26(0.53) | 6.04(0.81) | 3.88(0.61) | 6.54(1.10) |
| | Oracle | 3.49(0.25) | 7.21(0.50) | 11.45(0.57) | 27.48(2.14) | 2.94(0.27) | 4.66(0.57) | 3.43(0.54) | 5.94(1.08) |
| | SRDI(ours) | **3.43(0.34)** | **6.33(0.42)** | **11.55(1.17)** | **27.66(1.71)** | **2.65(0.46)** | **4.07(0.63)** | **3.28(0.51)** | **5.60(0.97)** |
| | *Improved* | +24.45% | +28.88% | +37.80% | +28.08% | +37.79% | +32.62% | +15.46% | +14.37% |
| ASTGCN | Partial | 5.57(0.72) | 10.61(1.36) | 22.44(1.58) | 43.07(2.46) | 6.14(1.29) | 8.95(2.35) | 3.60(0.60) | 6.05(1.13) |
| | Oracle | 5.04(0.39) | 9.59(0.62) | 19.17(0.91) | 40.21(2.02) | 4.54(0.47) | 6.48(0.85) | 3.47(0.50) | 5.83(0.99) |
| | SRDI(ours) | **4.45(0.43)** | **8.52(0.48)** | **21.93(1.13)** | **39.94(1.68)** | **4.56(0.63)** | **6.74(0.94)** | **2.96(0.46)** | **5.00(0.91)** |
| | *Improved* | +20.11% | +19.70% | +2.27% | +7.27% | +25.73% | +24.69% | +17.78% | +17.36% |
| MSTGCN | Partial | 4.78(0.43) | 9.35(0.75) | 18.96(1.21) | 40.13(2.67) | 4.75(0.73) | 7.02(1.42) | 4.43(0.87) | 7.61(1.86) |
| | Oracle | 4.49(0.31) | 8.93(0.50) | 17.41(0.74) | 37.84(1.88) | 3.64(0.41) | 5.60(0.82) | 3.39(0.52) | 5.82(1.06) |
| | SRDI(ours) | **4.22(0.45)** | **7.57(0.68)** | **17.29(1.10)** | **34.26(2.64)** | **3.67(0.55)** | **4.86(0.63)** | **3.25(0.26)** | **5.72(0.43)** |
| | *Improved* | +11.72% | +19.04% | +8.81% | +14.63% | +22.74% | +30.77% | +26.64% | +24.84% |
| T-GCN | Partial | 9.92(0.75) | 15.66(0.94) | 43.43(1.89) | 68.72(2.90) | 8.76(0.87) | 12.15(1.63) | 6.22(1.37) | 9.91(2.27) |
| | Oracle | 8.57(0.92) | 14.78(1.27) | 30.09(1.32) | 53.58(2.62) | 4.56(0.78) | 7.32(1.64) | 6.16(1.29) | 9.84(2.20) |
| | SRDI(ours) | **7.32(0.86)** | **11.42(1.01)** | **43.16(1.21)** | **64.44(1.67)** | **4.84(0.67)** | **8.13(1.10)** | **5.25(0.70)** | **8.31(1.07)** |
| | *Improved* | +26.21% | +27.08% | +0.62% | +6.23% | +44.75% | +33.09% | +15.59% | +16.15% |

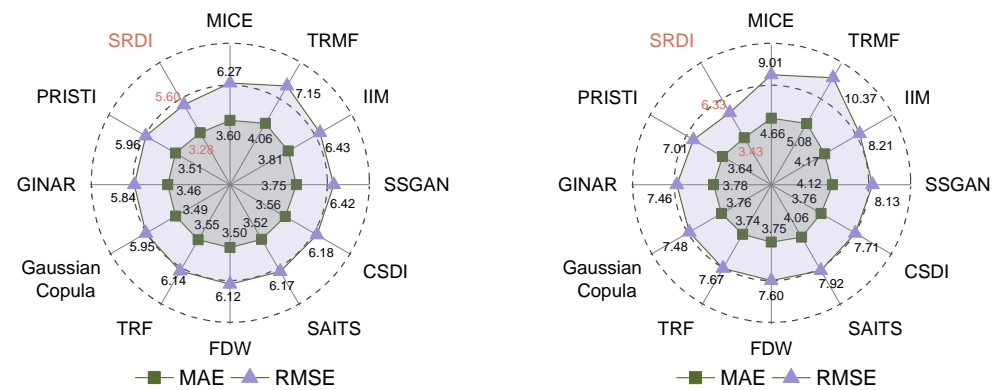

Figure 2: Performance comparison of imputation models on ECG5000.

Figure 3: Performance comparison of imputation models on METR-LA.

et al. (2024) , Gaussian Copula Zhao & Udell (2020) and SAITS Du et al. (2023). More information regarding the baselines can be found in A.3. Due to space limitations, we only present the experimental results on the ECG5000 and METR-LA datasets using MTGNN as the backbone here. Additional experimental results can be found in B.1. As shown in Figure 2, 3, our model achieves the best performances. These results highlight the limitations of current imputation methods in addressing the complete-missing variable problem for the VSF task. In contrast, SRDI effectively addresses the distribution shift issue in scenarios with missing variables, resulting in superior performance.

## 6.3 ABLATION STUDY

To validate the positive impact of each module in our model, we conducted ablation experiments. Due to space constraints, we present only the results for the ECG5000 dataset here.

**Spatial and Temporal Characteristics Preservation.** To validate the effectiveness of our innovations in the extraction of spatiotemporal features, we designed the following experiments: **1) SRDI-TS:** This configuration removes the Temporal Relation Extraction Module and the Global-Local Attention Adaptive GCN, replacing them with a linear layer. **2) SRDI-T:** This setup removes the Temporal Relation Extraction Module, focusing solely on learning spatial features. **3) SRDI-S:** In this version, the Global-Local Attention Adaptive GCN is removed, concentrating only on learn-

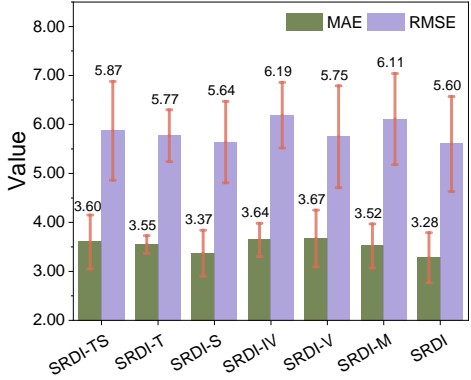

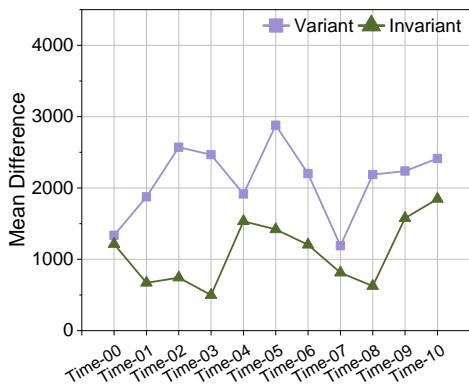

Figure 4: Performance comparison between SRDI and its various model variants on the ECG5000 dataset.

Figure 5: Comparison of inter-series correlation fluctuations between invariant and variant patterns on the ECG5000 dataset.

ing temporal features. The experimental results presented in Figure 4 indicate that SRDI-T, SRDI-S, and SRDI-ST all perform worse than SRDI, leading to the conclusion that extracting spatio-temporal features from time series is critical for missing variable imputation.

**The rationale and utilization of invariant and variant patterns.** To validate the effectiveness of our innovation in decomposing time series into invariant and variant patterns, we designed the following ablation experiments: **1) SRDI-IV:** This configuration eliminates the Series Invariant-Variant Dispatcher component, directly feeding the input into the Temporal Relation Extraction Module. **2) SRDI-V:** This configuration excludes the influence of the variant pattern, utilizing only the invariant pattern for the denoising process. Figure 4 shows that SRDI-IV underperforms compared to SRDI, confirming the effectiveness of decomposing time series into invariant and variant patterns for separate imputation, as they follow distinct dynamics. Additionally, SRDI-V performs worse than SRDI, indicating that the variant pattern carries important information that cannot be ignored.

**Meta-learning Strategy Against Intra-series Shift.** To demonstrate the effectiveness of our innovation in addressing intra-series shift using the meta-learning framework, we designed the following ablation experiment: **SRDI-M:** We removed the inner and outer loop training structure of meta-learning, reorganized them into a pipeline process, and eliminated adaptation during the testing phase. Figure 4 indicates that SRDI-M underperforms compared to SRDI, highlighting the effectiveness of our meta-learning framework in addressing intra-series shift and enhancing accuracy.

### 6.4 DEMONSTRATION OF THE DISPATCHER'S EFFECTIVENESS

To verify the effectiveness of the Series Invariant-Variant Dispatcher in distinguishing invariant from variant patterns, we conducted a visualization experiment. We computed adjacency matrices for both patterns at each time point, and for all but the first, calculated the difference between the current and previous matrices to capture inter-series correlation changes. To reduce randomness, we selected 10 samples, averaged the results, and plotted the differences as line graphs (Figure 5). The variant pattern showed more significant fluctuations, confirming the dispatcher's ability to distinguish between patterns. Visualization was limited to the ECG5000 dataset due to space constraints. Further results are available in B.2.

## 7 CONCLUSION

In this paper, we propose a Shift-Resilient Diffusive Imputation (SRDI) model for improving VSF performance by resolving distribution shift. Specifically, we classify the shift in VSF into two types: inter-series shift and intra-series shift. SRDI, a novel diffusion model-based approach to address the VSF problem, employs a divide-and-conquer strategy to tackle inter-series shift and enhances the meta-learning framework to address intra-series shift. Extensive experiments on four real-world datasets demonstrated that SRDI outperforms state-of-the-art methods, highlighting its effectiveness in addressing the distribution shift challenge in VSF tasks.

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

# A EXPERIMENTAL DETAILS

## A.1 METRIC DETAILS

This paper employs the metrics of two commonly used evaluation models: MAE (Mean Absolute Error) and RMSE (Root Mean Square Error). Their formulas are as follows:

$$\text{MAE} = \frac{1}{n} \sum_{i=1}^{n} |y_i - \hat{y}_i|$$

$$\text{RMSE} = \sqrt{\frac{1}{n} \sum_{i=1}^{n} (y_i - \hat{y}_i)^2}$$

where $y_i$ represents the true values, $\hat{y}_i$ represents the predicted values, $n$ is the number of data points.

Our model is evaluated in a partial setting. Therefore, we provide the improvement ratio of accuracy under the partial setting, as defined by the following formula:

$$Improved = \frac{ER_{partial} - ER_{SRDI}}{ER_{partial}} \times 100\%$$

where $ER$ denotes the error (MAE, RMSE).

## A.2 BACKBONES DETAILS

The basic introduction and implement details of the backbone models we use are shown as follows:

- *MTGNN* leverages a graph learning module to capture the uni-directed relationships among temporal variables and models the spatial and temporal dependencies using an innovative mix-hop propagation layer and a dilated inception layer. By integrating graph learning, graph convolution, and temporal convolution modules, the model excels in multivariate time-series forecasting by effectively capturing the correlations between time series data. In our experiments, we set the hyperparameters to match those used in the original paper.

- *ASTGCN* consists of three components that leverage spatial-temporal attention and convolution to model the three dynamic temporal characteristics of traffic flow. These features are then weighted and fused to produce the final prediction results. For the hyperparameter configuration, we used the same settings provided in the original paper.

- *MSTGCN* is a deep learning framework designed for modeling spatiotemporal data, leveraging multi-scale graph convolutional and temporal convolutional operations to effectively capture complex dependencies across different time scales, demonstrating superior performance in tasks such as traffic flow prediction and multi-object tracking. In our experiments, we utilized the same parameters as those specified in the original paper.

- *T-GCN* integrates graph convolutional networks (GCN) with gated recurrent units (GRU) to learn intricate topological structures and temporal data, enabling the capture of spatial-temporal dependencies. In our experiments, we utilized the same parameters as those specified in the original paper.

## A.3 BASELINE MODELS DETAILS

In this paper, we compare the proposed model with eight existing state-of-the-art imputation models. Below is a detailed introduction to these eight models:

- *MICE* imputes missing data by using a variable-by-variable approach through conditional densities. It iterates over these conditional densities, making it flexible for complex, multivariate datasets. The key advantage of MICE is that it doesn't require a suitable multivariate distribution like joint modeling (JM) and is effective when no single multivariate distribution can describe the data. Additionally, MICE's iterative approach, requiring relatively few iterations, allows for efficient and practical imputation.

- *IIM* addresses the challenges of missing numerical values by leveraging individual regression models tailored for each complete tuple and its neighbors. This approach tackles the sparsity problem by utilizing regression results from complete neighbors instead of their direct values, thus improving imputation accuracy. Additionally, IIM adaptively determines the number of neighbors for learning individual models to mitigate overfitting or underfitting, leading to more effective imputation outcomes compared to traditional methods.

- *TRMF* employs a novel autoregressive temporal regularizer to capture the structure of temporal dependencies among latent temporal embeddings in high-dimensional time series data. This method enhances the ability to forecast future values while effectively managing missing data. Its scalable design allows TRMF to handle large datasets efficiently, outperforming traditional time series methods that struggle with high dimensionality and noise.

- *CSDI* utilizes score-based diffusion models conditioned on observed data to handle time series imputation. The model is explicitly designed for imputation and leverages correlations between observed values to generate missing data from noise. Its advantages include handling both probabilistic and deterministic imputation tasks while improving accuracy compared to traditional methods, and it can also be applied to time series interpolation and forecasting.

- *FDW* is a method designed for handling missing variables in multivariate time series forecasting (MTSF). It works by retrieving nearest neighbors based on the available subset of variables and using them to fill in the missing values. The technique introduces a novel ensemble weighting method to handle the bias introduced by the partial dimensions during neighbor retrieval. The key advantage of FDW is that it can significantly recover forecast performance without retraining the underlying models, making it versatile and efficient in scenarios with long-term data loss or domain shifts.

- *SSGAN* is a method for imputing missing values in multivariate time series. It uses three components: a generator to estimate missing values, a discriminator to differentiate between observed and imputed data, and a classifier to predict labels and guide the generator. The method also incorporates a temporal reminder matrix to help the discriminator distinguish between real and imputed values. The key advantage of SSGAN is that it leverages both observed components and available labels, improving the imputation quality and ensuring accurate data distribution.

- *TRF* is a flow-based generative framework designed to impute missing variables in multivariate time series. TRF reconstructs missing variables by estimating the unknown conditional density of unavailable variables based on the available subset, using an invertible flow structure. This ensures accurate reconstruction by mapping the missing data to a Gaussian distribution and back. TRF's key advantage lies in its meta-learning framework, which allows it to generalize to different missing variable subsets without retraining, making it adaptable and efficient for dynamic real-world scenarios.

- *SAITS* imputes missing values by leveraging two diagonally-masked self-attention (DMSA) blocks, which capture both temporal dependencies and feature correlations between time steps. Its joint-optimization approach improves the imputation process by dynamically assigning weights to learned representations. The main advantage of SAITS is its ability to avoid the limitations of recurrent models, offering faster imputation with higher accuracy, and its non-autoregressive nature reduces the risk of compounding errors.

- *PRISTI* is a conditional diffusion framework for spatiotemporal imputation that enhances prior modeling by constructing and utilizing global spatiotemporal correlations and geographic relationships. It includes a conditional feature extraction module to capture effective spatiotemporal dependencies and a noise estimation module to transform random noise into realistic imputation values while mitigating the impact of added noise.

- *GINAR* is an end-to-end framework designed for multivariate time series forecasting (MTSF) with variable missing data. It leverages simple recursive units (SRU) enhanced with two key components: Interpolation Attention (IA) and Adaptive Graph Convolution

(AGCN). IA restores missing variables by generating plausible representations through attention mechanisms, addressing incorrect temporal dependencies. AGCN reconstructs spatial correlations between all variables, utilizing restored data to generate a reliable graph structure and improve spatial dependency modeling.

- *Gaussian Copula* model addresses the challenge of imputing missing values in mixed data (real, Boolean, and ordinal) by modeling the data as latent variables transformed through arbitrary marginals. Each variable—whether continuous or ordinal—is associated with a latent normal distribution, with ordinal levels represented as intervals. The model employs an efficient Expectation-Maximization (EM) algorithm to estimate copula parameters directly from incomplete data. This semiparametric approach ensures imputed values adhere to the statistical structure of the data, avoids the need for hyperparameter tuning.

## A.4 DATASETS DETAILS

- *METR-LA*
  This dataset comprises the average traffic speed data collected from 207 loop detectors installed along the highways in Los Angeles, covering the period from March 2012 to June 2012. The data is recorded at 5-minute intervals.

- *SOLAR:*
  This dataset includes solar power generation data from 137 solar plants situated in the state of Alabama, collected throughout the year 2007. The data is recorded at 10-minute intervals.

- *TRAFFIC:*
  This dataset contains road occupancy rates recorded by 862 sensors distributed throughout the San Francisco Bay area during 2015 and 2016. The data is recorded at 1-hour intervals. In accordance with Chauhan et al. (2022), an upscaling factor of $1e^3$ (multiplying the variable values by $1e^3$) has been applied.

- *ECG5000:*
  This dataset, obtained from the UCR Time-Series Classification Archive, consists of 140 electrocardiograms (ECGs), each with a length of 5000 data points, spanning a total duration of 20 hours. It is used for forecasting purposes, as illustrated in Cao et al. (2021).

## A.5 THE METHOD FOR COMPUTING THE ADJACENCY MATRIX

To express the relationships between variables, we computed a adjacency matrix between the variables. The commonly used cosine similarity was selected as the metric for measuring the correlation. The formula is as follows:

$$\text{Cosine Similarity}(\mathbf{X}^1, \mathbf{X}^2) = \frac{\mathbf{X}^1 \cdot \mathbf{X}^2}{\|\mathbf{X}^1\|\|\mathbf{X}^2\|}$$

where $\mathbf{X}^1$ and $\mathbf{X}^2$ are the time series of two variables. For $N$ variables, we calculated the correlation for each pair, and the resulting relationship matrix $C \in R^{N \times N}$.

## A.6 HYPERPARAMETER SETTINGS

We used identical hyperparameter settings for the ECG5000, SOLAR, and METR-LA datasets. However, due to the significantly higher number of variables in the TRAFFIC dataset, we specifically adjusted the embedding dimension of the diffusion model for this dataset.
- epochs: 100
- batch_size: 64
- lr: 1.0e-3
- block_number: 3
- itr_per_epoch: 1.0e+8
- dropout: 0.1
- layers: 1
- channels: 64

- nheads: 8
- beta_start: 0.001
- beta_end: 0.5
- num_steps: 1
- schedule: "quad"
- is_linear: False
- timeemb: 128
- featureemb: 16
- target_strategy: "random"
- diffusion_embedding_dim: 128(ECG5000, SOLAR, and METR-LA); 32(TRAFFIC)

# B SUPPLEMENTARY EXPERIMENTS

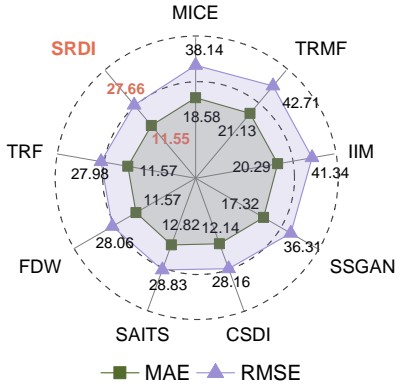 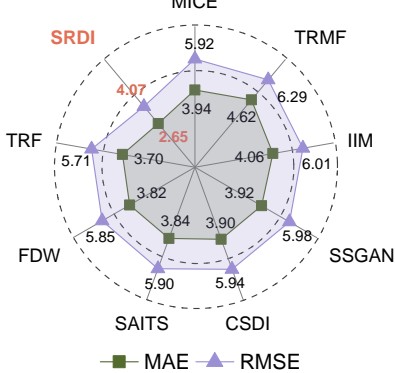

Figure 6: Performance comparison of imputation models on TRAFFIC.

Figure 7: Performance comparison of imputation models on SOLAR.

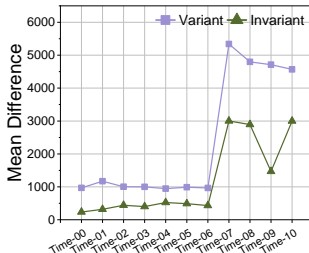 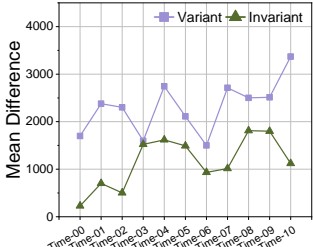 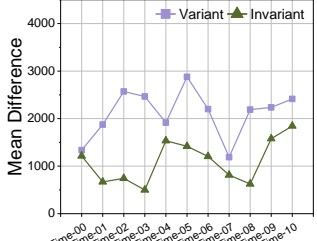

Figure 8: Comparison of inter-series correlation fluctuations between invariant and variant patterns on the METR-LA dataset.

Figure 9: Comparison of inter-series correlation fluctuations between invariant and variant patterns on the SO-LAR dataset.

Figure 10: Comparison of inter-series correlation fluctuations between invariant and variant patterns on the TRAF-FIC dataset.

## B.1 SUPPLEMENTARY COMPARISON WITH IMPUTATION METHODS

We have supplemented the experimental results comparing SRDI with baseline models on the TRAFFIC and SOLAR datasets. As shown in figure 6, 7, our model outperforms the state-of-the-art (SOTA). The experimental results across multiple datasets indicate that the effectiveness of SRDI is robust.

### B.2 SUPPLEMENTARY EXPERIMENTS FOR DEMONSTRATION OF THE DISPATCHER'S EFFECTIVENESS.

To validate that the designed Series Invariant-Variant Dispatcher effectively distinguishes between invariant and variant patterns, we conducted visualization experiments on additional datasets. The results are shown in Figure 8, 9, 10. From these results, it is evident that the Series Invariant-Variant Dispatcher successfully differentiates between invariant and variant patterns across all datasets.

### B.3 HYPERPARAMETER ANALYSIS EXPERIMENT

| weight | 0 | 0.0001 | 0.0005 | 0.001 | 0.01 | 0.1 | 0.3 |
|---|---|---|---|---|---|---|---|
| MAE | 3.23(0.73) | 3.28(0.51) | 3.01(0.73) | 3.12(0.62) | 3.59(0.64) | 3.28(0.56) | 3.61(0.71) |
| RMSE | 5.56(1.04) | 5.60(0.97) | 4.87(1.12) | 5.01(1.03) | 5.86(1.06) | 5.76(1.13) | 5.94(1.17) |

Table 2: Model performance under different weights of dispatcher loss.

$\varpi$ is a hyperparameter controlling the weight of the correlation disparity loss in the overall loss function. A small $\varpi$ may fail to distinguish invariant and variant patterns, while a large $\varpi$ could hinder diffusion model training. Additional experiments on $\varpi$ using the ECG5000 dataset with MTGNN as the backbone are shown in the table2. From the results in the table, it can be observed that the model performs best when $\varpi$ is set to 0.0005. As $\varpi$ decreases or increases from this value, the model's performance shows a declining trend.

| Model Name | Temporal Relation Extraction Module | Global-Local Attention Adaptive GCN | Series Invariant-Variant Dispatcher | Variant Pattern | Meta-learning Strategy | RMSE | MAE |
|---|---|---|---|---|---|---|---|
| SRDI | ✓ | ✓ | ✓ | ✓ | ✓ | 3.28(0.51) | 5.60(0.97) |
| SRDI-TS | ✗ | ✗ | ✓ | ✓ | ✓ | 3.60(0.55) | 5.87(1.01) |
| SRDI-T | ✗ | ✓ | ✓ | ✓ | ✓ | 3.55(0.18) | 5.77(0.53) |
| SRDI-S | ✓ | ✗ | ✓ | ✓ | ✓ | 3.37(0.47) | 5.64(0.83) |
| SRDI-IV | ✓ | ✓ | ✗ | ✓ | ✓ | 3.64(0.34) | 6.19(0.67) |
| SRDI-V | ✓ | ✓ | ✓ | ✗ | ✓ | 3.67(0.58) | 5.75(1.04) |
| SRDI-M | ✓ | ✓ | ✓ | ✓ | ✗ | 3.52(0.43) | 6.11(0.97) |

Table 3: Performance of models under different ablation settings.

## C TIME COMPLEXITY ANALYSIS

In the framework of meta-learning, we consider $K$ tasks, with each task involving one execution of the diffusion model and the forecasting model. The forecasting model serves as our backbone and is freely selectable. Therefore, when analyzing algorithmic complexity, we focus solely on our proposed model and exclude the forecasting model from consideration.

- For the diffusion model, the time complexity of the forward process is $O(R)$, where $R$ represents the number of diffusion steps. The backward process primarily depends on the design of our denoising model.

- Disentangling invariant-variant patterns requires computing a relationship matrix for each time step, resulting in a complexity of $O(T * N * N)$, where $T$ is the length of a time window, and $N$ is the number of variables.

- The temporal dynamic unit employs self-attention, with a time complexity of $O(N * T * T)$.

- The spatial dependency unit uses global-local attention, with a complexity of $O(N * T)$, and an adaptive GCN, with a complexity of $O(T * N * N)$.

In summary, the overall time complexity of our model is $O(max(K*R*N*T*T, K*R*N*N*T))$.

## D   ABLATION STUDY DETAILS

Table 3 provides a detailed description of the models used in the ablation experiments.. Removing any module results in SRDI losing certain critical capabilities, leading to a degradation in performance.

