# OpenReview forum: "SHIFT-RESILIENT DIFFUSIVE IMPUTATION FOR VARIABLE SUBSET FORECASTING"
_ICLR.cc/2025/Conference — Submitted to ICLR 2025_

### Official Review · Reviewer_kUPX · 2024-10-31

**Soundness:** 3
**Presentation:** 2
**Contribution:** 3
**Rating:** 3
**Confidence:** 3

**Summary:**

This paper studies the challenges of shifts in variable subset forecasting. It integrates a divide-conquer strategy with the denoising process and introduces a meta-learning paradigm, effectively addressing both intra-series and inter-series shifts. Experiments demonstrate superior performance against state-of-the-art methods.

**Strengths:**

1. This paper explores the issue of inter-series and intra-series shifts for variable subset forecasting.

2. This paper develops a divide-conquer denoising model for inter-series shift and meta-learning for intra-series shift.

3. Experimental results illustrate the effectiveness of the proposed method on four real-world datasets.

**Weaknesses:**

1. Several key claims require clarification. In section 4.2, why the divide-conquer strategy can handle the problem of inter-series shift?  In section 5, why did the author choose the meta-learning strategy to mitigate the intra-series shift?  The authors should explicitly compare and contrast the roles of Equations 11 and 13, and explain why both are necessary.

2. Two-stage training process may lead to error accumulation.

3. The presentation of core functional modules is mainly based on text description accompanied by snippets of equations that only show the intermediate transformation steps whereas the whole picture is lost. For example, it is not clear how Equation 11 and Equation 5 are constructed after reading the methodology section. The authors are supposed to provide a precise mathematical description of the entire data transformation process, i.e., how the inputs are transformed into the output step by step.

**Questions:**

1. The setting of the window length appears to be crucial for meta-learning. However, the paper lacks experimental analysis regarding its impact on forecasting performance.

2. Is the model easy to train? The authors should provide the complexity analysis of the proposed model.

3. The table number and title appear after the table. The references should follow a uniform style.

---

### Official Review · Reviewer_dqKu · 2024-11-01

**Soundness:** 3
**Presentation:** 2
**Contribution:** 2
**Rating:** 6
**Confidence:** 3

**Summary:**

## Overall Comments
This paper addresses the issue of missing data in time-series forecasting, which can lead to distribution shift challenges. To address this, the authors propose a novel framework called Shift-Resilient-Diffusive-Imputation (SRDI). The SRDI framework combines a divide-and-conquer strategy with a denoising process for more accurate missing data imputation. Additionally, the authors introduce a meta-learning-based training strategy for SRDI, which works in conjunction with backbone time-series forecasting models. Experiments conducted on various datasets validate the effectiveness of the proposed approach.

**Strengths:**

## Strengths
1. The paper's focus aligns well with the ICLR conference’s themes.
2. The module design is interesting, with attention to incorporating different neural architectures to enhance model performance.
3. The proposed training strategy is intuitive and provides a logical pathway to improve imputation accuracy.

**Weaknesses:**

## Weaknesses
1. **Technical Workflow**: The reviewer questions the suitability of directly applying diffusion models for missing data imputation. As noted in reference [1], *"the inherent mismatch in learning objectives between generation and imputation can impact imputation performance in diffusion models. Diversity, which is essential in generation tasks, conflicts with the imputation task's need for accuracy rather than variation."* Since diffusion models are sensitive to initial noise $x_T$, which facilitates diverse sample generation, the reviewer questions whether diffusion models are inherently suitable for precise imputation tasks.
2. **Divide-and-Conquer Strategy**: The “divide-and-conquer” strategy in Figure 1 appears to resemble the residual modules in the N-BEATS model [2]. How does this strategy differ in innovation from similar approaches such as N-BEATS?
3. **Missing Data Patterns**: Missing data patterns are typically categorized as missing at random, missing completely at random, and missing not at random [3]. Is the distribution shift primarily related to the “missing not at random” pattern? If so, it would seem crucial to incorporate the missing data mechanism during the modeling stage.
4. **Figure Reference**: Figure 1 is not referenced in the manuscript text.
5. **Experimental Design**: The sensitivity of the weight parameter $\overline{\omega}$ is not evaluated with respect to various strengths. Adding a sensitivity analysis for this parameter would enhance the experimental design.
6. **Experimental Presentation**: Error bars are absent in Figures 4 and 5, which would help demonstrate error bars in the experimental results.
7. **Typographical Errors**: For example, in the problem formulation, should $\mathcal{F}\_\theta$ be corrected to $\mathcal{F}\_\Theta$?

---

References:
[1] Self-Supervision Improves Diffusion Models for Tabular Data Imputation. CIKM 2024, Long Paper Track
[2] N-BEATS: Neural Basis Expansion Analysis for Interpretable Time Series Forecasting. ICLR 2020, Main Track
[3] Inference and missing data. Biometrika 1976

**Questions:**

## Questions
1. Could the authors provide a comparison with the ReMasker model [1]?

---
References:
[1] ReMasker: Imputing Tabular Data with Masked Autoencoding. ICLR 2024, Main Track

---

### Official Review · Reviewer_KDBw · 2024-11-02

**Soundness:** 1
**Presentation:** 2
**Contribution:** 1
**Rating:** 3
**Confidence:** 4

**Summary:**

This paper presents a Shift-Resilient Diffusive Imputation (SRDI) framework to address the Variable Subset Forecasting (VSF) problem - a challenging scenario where training data is complete but test data only contains a subset of variables due to sensor failures. The work identifies two key challenges in VSF: inter-series shift (changes in correlations between different series) and intra-series shift (distribution differences within the same series across time windows). The main technical contribution lies in combining a divide-and-conquer strategy with diffusion-based imputation. Specifically, SRDI decomposes the input into invariant and variant patterns to handle stable and fluctuating components of inter-series correlations. To address intra-series shift, the authors organize SRDI and the forecasting model into a meta-learning framework, treating time windows as tasks during training and incorporating an adaptation process for testing.

**Strengths:**

1. The paper provides a new view of using imputation-based approaches to solve VSF problem, particularly highlighting the limitations of existing methods in handling distribution shifts.
2. The proposed framework leveraging correlation matrices to explicitly model and capture invariant patterns in VSF, offering a more robust approach to handling inter-series relationships.
3. The integration of spatial information into the diffusion model architecture represents a significant methodological advancement, enabling better capture of complex spatiotemporal dependencies in multivariate time series.

**Weaknesses:**

1. Writing Issues:

1.1. Structural Inconsistency: While the paper claims to "categorize and provide a comprehensive analysis of two distinct types of shift in VSF tasks," this fundamental analysis is insufficiently developed in the main text. Such a core concept warrants a dedicated section rather than being merely mentioned in the contributions or inferred from conclusions.

1.2. Insufficient Technical Details: The paper omits crucial specifications regarding model parameters, training complexity, and inference costs. Given the framework's apparent computational intensity, these quantitative metrics are essential for evaluating its practical applicability.

1.3. Results Presentation: The experimental results should prioritize comparisons with other imputation methods and alternative VSF solutions, rather than focusing on various forecasting model combinations, given the two-stage nature of the proposed approach.

2. Experimental Concerns:

2.1. Counter-intuitive Results: The performance reported in Table 1, where the model surpasses the Oracle (full data) baseline with 85% missing data, raises significant concerns about the experimental setup or evaluation metrics. This counter-intuitive outcome requires thorough explanation or methodological reexamination.

2.2. Fairness in Evaluation: The meta-learning adaptation during testing introduces a fundamental fairness issue. The proposed method uniquely accesses and adapts to test data, while all other baselines (including Oracle) operate without such access. This creates an incomparable evaluation framework, making it impossible to fairly assess the model's actual contribution to VSF.

2.3. Limited Scope of Comparison: The paper focuses exclusively on imputation-based methods, overlooking significant non-imputation solutions (e.g., Chauhan 2022, GinAR from KDD2024). This limitation is particularly problematic given the fundamental differences between traditional imputation and VSF scenarios.

2.4. Incomplete Baseline Selection: The omission of recent relevant work (e.g., PriSTI 2023(mentioned in section 4.1)) from the comparative analysis compromises the evaluation's comprehensiveness.

**Questions:**

Please see above.

---

### Official Review · Reviewer_a52a · 2024-11-03

**Soundness:** 2
**Presentation:** 2
**Contribution:** 2
**Rating:** 3
**Confidence:** 4

**Summary:**

This paper propose a Shift-Resilient Diffusive Imputation (SRDI) model for improving the Variable Subset Forecasting (VSF) performance by resolving distribution shift. It classifies the shift in VSF into two types: inter-series shift and intra-series shift. Experiments on four real-world datasets demonstrated that SRDI outperforms several related methods in addressing the distribution shift challenge in VSF tasks.

**Strengths:**

1. The learning scenario of this paper is interesting.
2. Comparison experiments are conducted.
3. The source code is provided.

**Weaknesses:**

1.RELATED WORK is too simple. Many related studies are missing. The significant difference between the related studies and the proposed method of this paper is not analyzed, making the novelty of this paper be not clear.
2. The time complexity of the proposed algorighm is not analyzed.
3. The details of the adopted datasets are missing, such as the number of features and instances, the time interval between two instances.
4. The latest comparison model is MSTGCN Jia et al. (2021), more recent models are desired to be compared. Besides, some imputation methods could be compared, such as the methods introduced in Section 2.1 and “Missing Value Imputation for Mixed Data via Gaussian Copula, in KDD 2020”, “Transformed Distribution Matching for Missing Value Imputation, in ICML 2023”.
5. What is the impact of the hyperparameter ϖ.
6. Presentation issues. 1） line 124, page 3: In 2020, Lin et al. (2023) presented…, is it in 2023? 2) Figure 1 is not cited in the main text.

**Questions:**

1.RELATED WORK is too simple. Many related studies are missing. The significant difference between the related studies and the proposed method of this paper is not analyzed, making the novelty of this paper be not clear.
2. The time complexity of the proposed algorighm is not analyzed.
3. The details of the adopted datasets are missing, such as the number of features and instances, the time interval between two instances.
4. The latest comparison model is MSTGCN Jia et al. (2021), more recent models are desired to be compared. Besides, some imputation methods could be compared, such as the methods introduced in Section 2.1 and “Missing Value Imputation for Mixed Data via Gaussian Copula, in KDD 2020”, “Transformed Distribution Matching for Missing Value Imputation, in ICML 2023”.
5. What is the impact of the hyperparameter ϖ.
6. Presentation issues. 1） line 124, page 3: In 2020, Lin et al. (2023) presented…, is it in 2023? 2) Figure 1 is not cited in the main text.

---

### Meta-Review · Area_Chair_nVoe · 2024-12-20

**Metareview:**

This paper proposes a Shift-Resilient Diffusive Imputation (SRDI) model for improving Variable Subset Forecasting (VSF) performance by addressing distribution shifts, particularly focusing on inter-series and intra-series shifts. The paper considers an interesting learning scenario for forecasting under incomplete data. However, as raised by the reviewers, the novelty and the analysis of the proposed method are somewhat limited.

**Additional Comments On Reviewer Discussion:**

Although the paper has some merits, such as the interesting learning scenario, the issues raised by the reviews are critical. For instance, the related work section is too simplistic, lacking a thorough analysis of the novelty of the proposed method (a52a), the paper omits crucial technical details such as model parameters, training complexity, and inference costs (KDBw), and the suitability of diffusion models for precise imputation tasks is questioned (dqKu). Although the author addresses some issues in responses, the paper still needs a major revision before it can be accepted.

---

### Decision · Program_Chairs · 2025-01-22

Reject